# Students’ Perspectives on School Sports Trips in the Context of Participation and Democratic Education

**DOI:** 10.3390/children10040709

**Published:** 2023-04-11

**Authors:** Christoph Kreinbucher-Bekerle, Julia Mikosch

**Affiliations:** Institute of Human Movement Science, Sport and Health, University of Graz, 8010 Graz, Austria

**Keywords:** extracurricular activities, diversity, feedback, inclusion, physical activity, physical education (PE)

## Abstract

School sports trips, as a part of extracurricular physical education (PE), are a very important addition to regular PE, with benefits for not only physical activity behavior, but also for personal development and social inclusion. To better understand the relevance for students, the aim of this study was to look at their perspectives on school sports trips in terms of involvement, active participation, and co-designing opportunities. Therefore, 14 group interviews with 47 students (age: M = 13.9; SD = 0.9 years) were held in three exemplary secondary schools in Austria. The following six topics were derived from a qualitative text analysis: (a) the relevance for the students, (b) the motives for (non-) participation, (c) positive experiences, (d) barriers and challenges, (e) desired changes and ideas of the students, and (f) feedback opportunities. The results indicate that students are highly motivated to put forward their ideas for designing school sports trips in terms of physical activity and social components. This can further be considered for the planning and implementation of extracurricular PE, to make this an enjoyable experience for both students and teachers, promoting the relevance of physical activity in schools and beyond.

## 1. Introduction

Extracurricular activities in school play an important role, in addition to the regular physical education (PE), by including all types of physical activity, exercise, and sport in the educational context outside regular PE [1,2]. Overall, extracurricular activities can be categorized into those activities taking place inside the school facilities and those taking place outside the school facilities. For those activities taking place inside the school, there is physical activity during school breaks, either morning or afternoon [3], and voluntary activities in the afternoon, especially in all-day schools [4], and school sports days or sports festivities [5]. Outside the school facilities, there are excursions, including day trips or multi-day trips in summer or winter, as well as school sports events, either competitive or with a degree of play. In the German-speaking countries, these competitive events have a long tradition called “federal youth games” or “youth trains for the (Para-) Olympics” [6]. The school sports events taking place outside school facilities were the focus of this study, and are referred to as “school sports trips” throughout this article. 

Extracurricular physical activities, in general, are widely established in all types of schools throughout Europe [7,8]. They are implemented in schools and play an important role in integrating sport, exercise, and physical activity into daily school life [9]. These activities can act as a guide to prepare children to be physically active outside of school and have long-term positive impacts on physical activity, healthy behavior, and obesity in children [10]. Furthermore, these activities can contribute to the creation of healthy habits and can promote active lifestyles [11,12]. Extracurricular activities can also have positive effects in terms of the relationship between PE, sports, and social inclusion [13]. Regular participation in extracurricular activities can also lead to better emotional and behavioral adjustment and promote social competence [14]. In the growing concept of all-day schools, the incorporation of physical activity in the school day is very important since there is less time for the children during the week to attend sports events and to take part in physical activities outside of school [15]. This is even more alarming if we consider the current public health recommendations for physical activity [16] and the low amount of children meeting the criteria of at least 60 min per day of moderate-to-vigorous intensity physical activity (MVPA) and its relationship with other parameters, such as obesity, physical fitness, and screentime [17]. Moreover, it is important to try to offer pleasurable activities and as much as physical activity as possible in schools, particularly after the influence of COVID-19 restrictions, which did not allow extracurricular activities in this context [18,19], where severe effects were found on all levels of health and well-being. It was not only that students became less capable of motor skills and had a higher BMI [20], but there was also an impact on the mental health components related to (non-) participation in extracurricular activities [21].

One way to promote the positive effects of extracurricular activities and overcome health-related implications is by ensuring that the voices of students are heard regarding physical activity offerings in schools. In this sense, a concept which is gaining more attention in recent years in the educational context is participation and democratic education [22]. Based on the idea of democracy and education [23], the first attempts to connect PE to democratic values and participation were made 70 years ago [24], whereas sport and democracy have a long tradition [25]. In the last few years, participation and student orientation have gained more consideration around sports and PE in schools [26,27,28]. The concept of participation is sometimes difficult to grasp, as it does not only mean taking part, but also to be integrated and involved in the decision-making, and to be consulted, which is more in the direction of democratic education. The concept of participation and democratic education in PE or extracurricular activities is even more interesting, but also challenging when we consider current topics such as diversity and inclusion [29]. Besides findings regarding the inclusion of students with disabilities in PE [30,31], there have also been investigations of inclusive school sports trips and the participation of students with disabilities, for example in Austria [32,33,34], the United States [35,36], and Canada [37]. There has also been some consideration on how participation of students with disabilities in extracurricular activities is possible and what the role of people with disabilities is [38,39]. However, there are limited options for students with disabilities or special educational needs (SEN) with regard to the inclusion in extracurricular PE [40]. In this context, a current overview revealed that only 30% of children with additional needs were included in extracurricular PE [41]. There is room for improvement here, even if we do not look at disability itself, but rather the interconnectedness with other dimensions, such as age, gender, and migration, in terms of intersectionality [42]. Related to this, significant disparities in participation in school-based extracurricular sports and physical activities have been found among girls with disabilities [43].

By including the students’ perspective, meaningful and goal-oriented measures can be established that increase the participation opportunities for all students, and implications can be derived for the design of specific activities. One way to increase participation of different students in physical activities in schools is to ask students about their needs and perceptions [44], or to find similar ways of contribution in the sense of democratic education [29]. The mentioned findings indicate that it is unclear how all students are included and if their voices are heard with regard to school sports trips. In addition, some students seem to be less present than others, especially when considering students with disabilities [45] or those with a different cultural background who are potentially not familiar with the language [46]. To date, little is known about participation options, and asking about students’ needs has not been explored for school sports trips. Therefore, in this study, the students’ perspectives on school sports trips were investigated, addressing the following research questions: (a) which basic concepts do students understand about school sports trips and their participation, and (b) which opportunities do they have to be actively engaged in these activities?

## 2. Materials and Methods

### 2.1. Participants and Procedure

In 14 interview settings, 47 students (29 male and 18 female) from three exemplary secondary schools in Austria took part. These schools were from one region and were comparable in terms of their educational orientation. Students attending the seventh or eighth grade in six different classes were put forward by their teachers to participate in the interviews. This age group was used due to the fact that some offers of school sports trips start in the sixth grade. Informed consent of their parents was obtained for all participating students. The age range of the students was between 12 and 16 years (M = 13.9; SD = 0.9), and seven of the children interviewed (14.9%) had SEN. Eight of the 47 students (17.0%) were not born in Austria, and 19 (40.4%) had at least one parent who was not born in Austria. On a five-point Likert scale (with “1” being the best score), students rated PE with an average of *good* (M = 2.3; SD = 0.9). In total, 87.2% of the children were physically active outside of school, with football/soccer, cycling, and swimming as the most common types of physical activity.

The interviews were conducted at the end of the 2020/21 school year in small groups consisting of three to four students, to create a relaxed environment for the students and to enable discussion and exchange between them. If possible, the students were mixed and balanced on age, gender, SEN, and migration background in every interview situation (see Table 1). This format was already found to be helpful in a similar investigation, where students were asked about their perception of PE [44]. The benefits of a small group interview setting are the cost-effectiveness and the fact that this setting represents a safe peer environment for children [47], in which they might contribute more than in one-on-one interviews. However, this setting also creates some limitations, in that not all the questions were answered by all 47 students. 

### 2.2. Interview Design

A semi-structured interview design with guidelines was created to specifically find out the students’ points of view, their wishes, and their suggestions for change, as well as the problem areas and existing potentials based on empirical findings [1,21,33]. The guidelines were divided into the following main parts: (a) school sports trips and offers, (b) preparation for these activities, (c) participation opportunities, (d) feedback mechanisms, and (e) COVID-19—related influences. To illustrate the design and comprehensibility of the interview guidelines, the following are example questions for each part: (a) “Which extracurricular offers are present at your school?” (b) “How do you and the teachers prepare for a school sports trip?” (c) “Which opportunities do you have to make a choice or pursue your interests in school sports trips?” (d) “Which opportunities of giving feedback do you have in school sports trips?” (e) “Did the COVID-19 pandemic have any effects on your school sports trips?”. Following these introductory questions, the interviewer went into more detail with standardized as well as spontaneous questions.

### 2.3. Data Analysis

For the assessment, the interviews were recorded, transcribed, coded, structured, and analyzed according to qualitative text analysis [48] in a deductive–inductive procedure. We started with basic categories derived from research on participation [27], student orientation [28], and democratic education [29]. The following six topics were than derived from the analysis, in which two independent reviewers agreed on a consensual basis: Relevance for the students.Motives for (non-) participation.Positive experiences.Barriers and challenges.Desired changes and ideas of the students.Feedback opportunities.

Although the elementary approach of this investigation was a qualitative analysis, with regard to the considerable sample size required for a qualitative investigation and the nature of the group interviews, the aim was to obtain a broad overview of the students’ perspectives. Therefore, the analysis was quantified to show the distribution of the frequencies of students’ needs and to allow conclusions in terms of potential generalization. This quantitative rationale represented an attempt to increase the expressiveness of the results.

## 3. Results

The structure of the results is based on the two research questions and the six categories derived from the analysis, after a brief description of the offers attended by the students. In the quotations, the corresponding interview is indicated by a letter (A)–(N), and the number (1)–(4) serves to distinguish the students. A quote from M3 therefore means that the third student in interview session number 13 made this statement (the characteristics of all the students are summarized in Table 1).

### 3.1. Descriptive Data for Attended School Sports Trips

Specific activities were mentioned by 32 students (68.1%) who gave one or more answers for school sports trips they had attended. The most common activities were skiing (40.0%), hiking (21.8%), summer sports week (10.9%), tournaments (7.3%), sport days (6.3%), climbing (5.5%), swimming (3.6%), and other activities not directly related to physical activity, such as get-to-know days or city trips. 

### 3.2. Relevance for the Students 

The question on the personal relevance for the students was answered by 34 students (72.34%). The distribution of the different answers is shown in Figure 1. 

The most frequently mentioned answer in this category was a *change from everyday school life*, with 31.9%: “As motivation and reward for school, afterwards it is easier again.”–M3. “It is a bit like a holiday from school and relaxing.”–K3.

*Improving the class community* was mentioned in 29.8% of the answers: “I think something like that just increases the class community and you just get to know each other better.”–C1. “[…] that you grow better together with friends.”–A1.

*Great experiences and adventures* were mentioned in 23.4% of the answers: “We see things of the world that you don’t normally see and experience.”–M1. “Memories.”–D3.

For 12.8%, the focus was on *general fun*, and 6.4% cited *practicing their sport(s)* as of personal importance. “And the fun together and enjoying the time.”–D3. “[…] that you also do exercise, that’s always good for you.”–I3.

### 3.3. Motives for (Non-) Participation

A total of 29 students (59.2%) gave one or more answers to the question asking their motive as to why they attended certain trips. In total, 33.4% of the answers were about the fact that they can *spend time with friends and classmates*: “Mainly because of my friends.”–D3. “I think it improves the friendship because you’re in the same room and also the class community [improves].”–C3.

For around 22.2%, it was the *interest in sporting activity*: “[…] I just like skiing, I enjoy it.”–D1.

A further 17.8% of the answers had named *the special nature of the trips* as the background for these activities: “That you get out with the class for a change. I’m immediately more motivated than when I’m sitting in class doing maths. And then you just have a lot more fun.”–G1.

A total of 13.3% of the responses addressed the *fun* of such events: “Because it will definitely be fun.”–N3. Others expressed a sense of *obligation* to be at such events, as explained by D4: “We have to go.” 

The following answers either refer to the students themselves (their own experiences) or they show possible reasons that they have heard from others that have spoken against participation in school sporting events. There were 39 answers to this topic, with some students answering more than once. Thirty students (63.8%) answered this question in total, and 17 (36.2%) gave no answer. 

With 25.6% of the responses, *lack of interest* in sports is the most common reason for staying home or not participating: “Maybe [because] they just don’t want to or [because] they don’t like skiing.”–A2. “I just don’t like it, the sport, skiing.”–L4.

For 15.4% of the responses, both *lack of ability* and *financial reasons* were cited as the second and third most common reasons for non-participation in school sporting events: “I think most of them just don’t want to or can’t do the sport.”–M2. “Financially it’s certainly an issue for many, for every trip, small or big, we have to pay something.”–G2.

A total of 12.8% of the statements referred to *problems with the class community* and *homesickness*: “Well, if you don’t like doing something with the class, so you don’t feel so comfortable there.”–B2. “Some probably also because of homesickness.”–M1. The lack of permission from parents was mentioned as a reason in 7.7% of the statements.

For 5.2% each, non-participation was attributed to a lack of permission due to *misbehavior* or *illness* and/or a *restriction* of the opportunity to participate: “You (at F3) almost weren’t allowed to go!”–F1. “Right, I was a bit too bad before that, but then I was still able to [ride].”–F3. “So I know about one girl, she didn’t go because she has an illness and [I] think a boy from our class didn’t go, but I don’t know why.”–H3.

### 3.4. Positive Experiences

Of the 47 students, 39 (82.9%) were able to name one or more positive things that they remembered or liked about the school sports trips, while the remaining eight (17.1%) could not remember anything or had no good experiences with school sports events.

For 31.9% of the students, the relevant *sporting activity* was in the foreground: “I thought it was great that there was a snowboard group.”–C1. “The skiing itself.”–D2. “Yes, the basketball tournament.”–J3.

A total of 17.0% of the statements mentioned the activities as a *nice, fun, and exciting time*, without wanting to commit to anything specific. These statements were grouped under the category of general fun: “Everything actually, I really liked it.”–N3.

*Other activities* that were additionally undertaken at school sports trips, such as torchlight hikes, games evenings, or a disco at the ski course, were a particularly positive aspect in 14.9% of the cases: “The torchlight hike.”–N1. “I think the disco was cool.”–N2.

A total of 8.5% of the statements mentioned the *time spent together* with the class and/or time with friends as a positive memory, and 8.5% of the answers told of a special and welcome change, through school sports trips, from the stressful everyday school life: “Yes exactly, not being in school for a change, but doing something outside with the class.”–C3.

### 3.5. Barriers and Challenges

Of the 47 students, 20 (42.6%) mentioned one or more situations they were dissatisfied with or unhappy about on school sports trips. Barriers and challenges related to *restrictions* with the *COVID-19 pandemic* was a topic for 31.9% of the students. Students mentioned the uncertainty in this regard: “At the beginning we were told that we would make up for it, for sure, but soon it was clear that it was totally cancelled and that was of course frustrating.”–B1. They were also very frustrated that these events could not take place: “It was really stupid to take even that away from us.”–C3. They also critically remarked that they would have liked to have been offered something instead of the joint activity, as the following statements show: “Baking pizza online or watching a film together or something.”–H2. “A film evening as a class and something as a community would have been cool, because that has also been totally dropped.”–B2.

For 17.4% of the students, the sport and physical activity was clearly *too strenuous*, and they had lost the fun of it as a result: “For me, it was just that the skiing course was usually very strenuous, but the teachers thought we were simply overdoing it because we didn’t want any more or something and didn’t notice that we were […] at our limits. And after the five days we were totally exhausted.”–M2.

Another 17.4% were *dissatisfied* with the *accommodation* and/or *food* on site: “The hostel was catastrophic, everything smeared, and the food was not good and very little.”–K1. “So once when skiing, someone had to sleep on a folding bed that was folded up all the time.”–I1.

For 8.7%, the *overall situation* was negative. This was associated with many different factors. These students did not mention any positive experiences in the interview, or the challenges outweighed the positive experiences in their eyes: “The ski course was not mine at all. I really like skiing and usually go on skiing holidays with my parents every year and so on, but the ski week there was just super unpleasant from the place and not a nice week for me. The skiing was really good, but the downhill was stupid.”–H1.

A total of 8.7% were unhappy with the *arrival situation* as they had to travel extra every day for their ski course (without having accommodation on site): “The other class got to sleep up-stairs, we went up and around every day.”–F1. “And why?”–[Asked by the interviewer]. “Because how the previous third [classes] performed, we weren’t allowed to stay overnight.”–F1.

A further 8.7% of the statements mentioned dissatisfaction with the *group allocation*, either for the accommodation or the type of activity they performed together, which was set for the practice of the activities.

### 3.6. Desired Changes and Ideas of the Students

From 31 students (65.9%) came one or more ideas as to what they would like to see changed in the future. The most frequent suggestions (42.6%) were for *more sports or types of excursions*, including climbing, swimming, golfing, and cycling: “I would like to go to the seaside, you can do sports there too.”–B2. “Cycling.”–F1. “Yes, climbing would be cool too!”–L3. “If I could choose anything it would be golfing or trying mini golf.”–M2. “Wind tunnels, where you fly, [I] think they have that here.”–N1.

In 14.8% of the cases, the topics of *including student opinions* more or voting on topics were mentioned together: “In any case, our opinion doesn’t really count for anything, or no one is seriously interested, kind of a pity.”–H2. “I think the students should really be allowed to have more of a say, and if we were asked, we would certainly get more involved.”–E3. “Yes, by talking to the students beforehand, maybe. So that you don’t have to choose between bad and even worse (laughs).”–C2.

In total, 11.7% of the statements dealt with *rules* that were *too strict*; some mentioned the topic of too little free time: “Maybe more free time, that would come to my mind.”–A1. Others mentioned rules prohibiting mobile phones that were too strict, and the implementation of a point system: “Most likely the ban on mobile phones, most students do it secretly anyway, and I understand it anyway besides skiing […] but in general it’s a bit unnecessary. And then there was also a point system, what our room looked like. And if you have someone in your room who is untidy, it ruins everything. Above all, we’re talking about one week! The teachers know who’s messy anyway and that was really just childish.”–B1.

A total of 8.2% of the students’ statements stated that they would like to have *more involvement in the planning*: “Yes, one or two days, staying overnight doesn’t have to be, but that makes it cooler and special.”–C2.

### 3.7. Feedback Opportunities

One part of the interview was aimed at surveying the current state of feedback, in terms of if the students were asked for feedback after school sports trips or if this possibility even exists at all. The evaluation was based on the individual interviews, as, in most cases, only one of the three or four students answered this question. Out of 14 interviews, feedback was asked for in four cases (28.6%), but on two occasions, this was not implemented: “They [do] ask, but they don’t change it.”–H1.

In two interviews (14.3%), this question was not answered at all, and, in the remaining eight interviews (57.1%), it was answered, for example, as follows: “No, not really, but actually it would be cool to get a questionnaire after the week to give feedback then.”–A1. Or “No [unanimously].”–D1, D2, and D3.

The question of whether the students would like to be able to give feedback after school sports trips was answered by 42 students (91.5%). A total of 35 students (81.4%) would like to be able to give feedback: “I think students should really be allowed to have more of a say, and if we were asked, we would certainly be more involved.”–E3. “Yes right away. Totally.”–A1 and A2. “If someone asks me, I would already like to give my opinion and bring ideas, but you really get the feeling that they don’t care anyway.”–H3. “Would make sense, the teachers could then understand us students better and do it better next time.”–M2.

A total of 14.3% of the students did not think it was important or did not want to give feedback: “[I] don’t think the feedback is very useful.”–Q3. “No.”–G1.

In total, 4.8% of the respondents would only give feedback if they could also see their actual change and implementation.

## 4. Discussion

The current investigation examined the students’ perspectives on school sports trips, incorporating all physical-activity-related activities which take place outside school facilities (hourly, daily, or multi-day activities). According to the answers of the students, the topic of school sports trips holds a lot of meaning for them. These school sports activities are often remembered by the students and can be a great opportunity and supplement to regular PE if they are properly timed, planned, implemented, and evaluated. Basically, the activities of the different schools in Austria look very similar. For the weekly trips, most of them offer ski courses and summer sports weeks, if not cancelled due to the restrictions during the COVID-19 pandemic. Otherwise, there are hiking days and smaller excursions (swimming, climbing, etc.). This is comparable to the data of a questionnaire for Austrian students in which summer sports were mentioned as the most favorable for them [34]. The offer of leagues, competitions, and day events was only rarely mentioned by the students in the current investigation. Participation offers may also differ in terms of gender, educational, or migration background [49], to which the current data cannot contribute. Contrary to the expectations drawn in the introduction, diversity and inclusion was not made a specific topic for the students. 

According to the first research question regarding the concept of school sports trips, this seems, in general, to be highly relevant for the students. Most often the school sports trips were seen as a welcome change from the daily school life and an additional experience. Another mentioned point in the context of relevance was improving the class community, friendships, and having fun together. These social (inclusive) effects of extracurricular PE are very important and common [50]. For many of the students, these events are primarily associated with the social component, and thus social cohesion is a relevant factor, which is not only important in the general educational context [51], but also in school sports trips [50]. The social component of school sports trips was also dominant in the positive experiences, in which students mentioned things that happened during the event, in addition to the physical activity itself, such as games or other social events. The social aspects, such as the time with their friends and the class community, and the peripheral activities are very important to the students, and these points should not be underestimated in the planning. A stable class community and good friendships can therefore play a major role in school sports events and are a prerequisite for many students to participate in them. Spending time with classmates and friends was also mentioned most often in the motives for taking part in school sports events. 

It is surprising that the motives and motivations or the personal relevance of these events for most of the students are not only related to the sport or physical activity itself. Designing such events specifically could be a good motivation for all students, irrespective of their interest in sports or physical activity. In this sense, discussing the benefits of school sports trips next to PE should be relevant, as there may well be physical, social, affective, and cognitive effects. In the social domain, there is sufficient evidence to support claims of positive benefits for young people. Importantly, benefits are mediated by environmental and contextual factors such as leadership, the involvement of young people in decision-making, an emphasis on social relationships, and an explicit focus on learning processes [52]. Therefore, a wide range of optional activities should be created, which also loosens up the framework conditions and increases the probability that there is something for every student. In addition, the students should have at least a certain amount of co-decision-making power. To increase this even more, the teacher could ask in advance what activities the students would like to perform and what is disturbing and/or relevant for them, to use this as a foundation for further planning. In the last few years, co-creation is obtaining more and more attention in the context of physical activity [53], and also in the educational context, e.g., in PE [54]. It is also a good way to increase the participation of underrepresented groups or people who are not as physically active as others [55,56]. There is also evidence of the positive effects of cooperative planning in the sense of health promotion [57], as well as in many domains related to physical, social, and psychological variables [58]. The nature of school sports trips can contribute to all this, which is especially important with the limited number of hours of regular PE. 

Looking at the values of the students’ statements, it is clear to see that they are very motivated in their own eyes, especially among themselves. Another reason is the interest in the physical activity, which can be seen as two-fold. One the one hand, students who are highly interested in sports and physical activity very much look forward to these events, but the others who are not that interested are questionable. This also shows that a lack of interest in sports and a lack of ability are the commonest reasons for not taking part in these events, followed by financial reasons. Another reason was lack of friendship or homesickness. Thus, all the motives in the positive direction are also barriers in the other direction. Some of these factors cannot be directly influenced by teachers and assistants. However, motivating and committed (PE) teachers can inspire their students and prepare them in the best possible way for such events. In this way, “lack of motivation” could most likely be tackled directly by the teaching staff and the school. This can also be connected to the pedagogic quality of school sports trips, e.g., the attention to concepts such as student orientation or incorporating social values [28]. Thus, educational institutions in teacher education should not only address the design of specific offerings in extracurricular PE, but should also consider the needs of the different students and implement teaching principles during and in preparation of these activities. However, it should be taken into account that, besides PE teachers, other support staff members are also likely to be taking part, including teachers without qualifications in sports or trainers without school-related education [59]. To ensure a pedagogical quality which is more student-oriented could be one way to increase the intrinsic motivation of students [60]. In addition, it is also important to consider the relevance of extracurricular PE in the curriculum or other binding documents [61], and a discussion of PE and extracurricular PE in terms of health-related educational and pedagogical dimensions should be conducted [62].

Regarding the second research question about involvement opportunities for students and their active engagement, a great demand of the students was for their opinion to be included even more. This was mentioned not only related to the trips themselves, but also to the planning process of an activity, which could be used to increase the number of students taking part or the likelihood of having enjoyable moments together. Many students had ideas on how a program could be strengthened or improved. Their interests should be considered, because, with the improvement and expansion of the activity, different interests are also covered, and the motivation of the students might increase. Despite their ideas, students were also skeptical about feedback opportunities related to school sports trips. For them, giving feedback seems very important, but they have the feeling it is not heard that often [63]. As perspectives and preferences on feedback can differ considerably, it seems to be advisable for the teachers to discuss with their class which method to choose before conducting the feedback, or to cover several feedback methods. Especially with feedback, students should not have to fear offending someone with honest information or negative consequences. This commitment should be explained to all students beforehand and then kept. Feedback and participation opportunities should be introduced for all students, irrespective of their background or potential SEN [64]. It might also be important to install basic rules for giving feedback in schools [65]. A little bit is known about how students are dealing with feedback [66], but so far nothing is known about how it is implemented in school sports trips, keeping in mind that giving feedback is a form of participation, involvement, and democratic education. However, it should be clear that students are more comfortable with some methods than with others. Even though most of the students interviewed were in favor of an oral feedback method, this does not have to apply to all learning groups or students. Possible ways to enhance opportunities for students would be a questionnaire or an anonymous feedback box. Other forms of feedback are related to increasing student participation, such as having class representatives and being able to formulate democratic decisions [22,67]. Extracurricular activities in schools can be a good field for democratic education because they offer more room for flexibility and are not directly connected to obligations related to the curriculum [2]. Especially in daily and multi-day school sports trips, implementation should be easily possible with the above-mentioned feedback box or discussion rounds, e.g., during travel or for a get-together after the sport and physical activity. Giving students a voice and increasing participation opportunities will not only help to improve the quality of the activities, but will also benefit the general health and well-being of all students in schools [68,69]. 

There were a few things with which the students were unhappy. Firstly, the strenuous activity and the focus on performing a lot of sports and physical activities. According to the students, there could be more flexibility in the activities. Other issues were with the accommodation and the food on site, which is improvable, according to the students. There were also a few students who mentioned that they just did not like school sports trips at all. As expected, the restrictions around COVID-19 were also a limiting factor, and one can see the desire in the interviews of many students and the wish to participate in such events again. Moreover, the wish to be more flexible in thinking and finding other (online) opportunities to come together, in order to not lose the valuable social components of these events, was present. Empirical findings related to COVID-19 suggest that it was difficult for the teachers to keep up with the daily business and the implementation of online learning in schools [70], whereas other activities could not take place due to restrictions or inflexibility. Potential effects go along with mental health aspects and the well-being of students [21]. 

Finally, desired changes or ideas of the students were related to other sports and physical activities. This could be a chance for schools to expand their activities in the context of physical activity and to get in touch with external providers and trainers. Therefore, it is important to connect with voluntary sports clubs or to cooperate closely with sports organizations to increase the likelihood of exciting physical activity offers [71]. All of these ideas are important to obtain more active school communities, in general, and could be a predictor for sports participation outside school [11]. 

## 5. Limitations

Due to the nature and procedure of the group interviews, they did not allow an in-depth analysis. Moreover, a group interview setting can have the limitation of not everybody participating, e.g., those struggling with language or speaking. It is possible that some of the students would have said more in an individual interview situation, but it could also be the case that they would not have felt confident in this setting with a researcher they did not know and without their classmates. Therefore, this was a good opportunity to take part in an interview with others, and it seems to be a suitable approach for a first indication on the topic, with a procedure that has been proven to be valid in similar investigations [44]. Considering the potential methodological limitations, further qualitative investigations are recommended. 

In the introduction, we addressed the manifold effects of extracurricular physical activities in schools, such as enhancing health-related physical activity, personal development, and social inclusion. Some of these factors were also named by the sample in the current investigation, which gives a first overview on the students’ perspectives. By further drawing on this idea, it would be interesting to research the interaction of the potential effects of school sports trips.

Furthermore, the results cannot be generalized because it was a sample selected by their respective teachers, taken in one region of Austria. The subjective selection could have led to the teachers choosing students that were highly interested in the topic. However, although the sample might be representative for the situation in Austria, in other countries, extracurricular physical activities will be different, and participation opportunities may differ [7]. Still, the option for students to participate and have their choices heard should be a human right, no matter where someone is born or lives [72]. 

## 6. Conclusions

The current investigation suggests that the topic of school sports trips is very important for students, and they have a lot of ideas that they want to share. Extracurricular activities can be a good basis to increase students’ involvement and participation in terms of democratic education. In general, school sports trips have great potential in various effects, but more empirical data on extracurricular PE and participation options of students, irrespective of their background, are needed to address the needs of the students, ensure the pedagogical quality, and enable more enjoyable moments in various physical activity contexts in schools. 

## Figures and Tables

**Figure 1 children-10-00709-f001:**
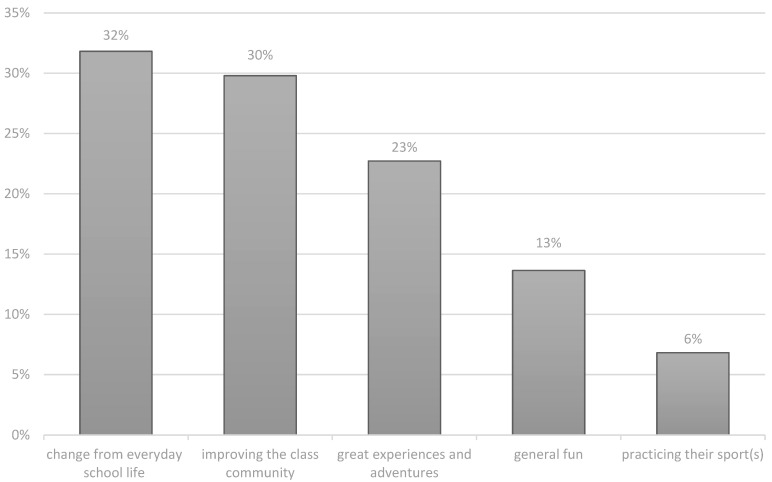
Overview of the students’ answers to the topic of the personal relevance of school sports trips.

**Table 1 children-10-00709-t001:** Overview of the participating students and their sociodemographic data, according to the interviews.

Student	Age	Gender	Migration Background	SEN
A1	14	M		
A2	15	M		
A3	14	M		
B1	15	M		
B2	13	F		
B3	15	F	parent	X
C1	15	M		X
C2	15	M	parent	
C3	14	M		
D1	13	M		
D2	13	F		
D3	13	M		
D4	13	F		
E1	14	F	self	
E2	14	M	self	X
E3	12	F		X
F1	13	M	self	
F2	13	M		
F3	13	M		
G1	14	F	parent	X
G2	13	M		
G3	13	M		
H1	15	F	parent	
H2	13	F		
H3	14	F	parent	
I1	13	M		
I2	14	M		
I3	14	M	parent	
I4	13	F	parent	X
J1	14	F		
J2	14	M		
J3	14	M	parent	
J4	14	M	parent	
K1	15	M		
K2	16	M	self	
K3	14	M	parent	
L1	16	F		
L2	14	M		
L3	13	M		
L4	14	M		
M1	15	F	parent	
M2	14	F	self	
M3	14	M	self	
M4	14	M	self	X
N1	14	F		
N2	15	F	parent	
N3	15	F	self	

SEN = special educational needs. F = female. M = male. Note: The number of the interview is indicated by a letter (A)–(N), whereas the numbers (1)–(4) serve to distinguish the students in the interviews.

## Data Availability

Data and materials associated with the current study are available from the corresponding author on reasonable request.

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
