# Peer review of "Students’ Perspectives on School Sports Trips in the Context of Participation and Democratic Education"

_children, 2023, doi:10.3390/children10040709_

Round 1
Reviewer 1 Report
An interesting study that addresses a relevant issue. It is well written. My suggestions are to improve the rigour of the work. The first comment is that the data analysis strategy needs expanding. Qualitative benefits from being able to explore and identify new areas, yet the results are reported in frequency counts. I understand why this is the case, but it would be helpful if the authors presented a strategy. They are balancing a large sample for qualitative with representation.
The second point is that more details on who the participants were. People who volunteered to participate in this study, that is, with experience of after school activity might represent a specific sample. Children who avoid doing after school activities, for example, are not represented and yet, when start speaking of the health related benefits of doing after school activities, it is this group who would benefit the most. Rather than hide this issue in the article, my suggestion is that they express the sample as using people who volunteered and thus could have a positive bias from the outset.
In summary, strengthen the methods and this should enhance the work.
Author Response
We want to thank the reviewer for the many helpful and thoughtful suggestions on how to improve the manuscript. Based on those, we revised the methods and specified our analysis. Our adaptions (via track changes) will be explained according to the comments. We gladly addressed them as follows:
The first comment is that the data analysis strategy needs expanding. Qualitative benefits from being able to explore and identify new areas, yet the results are reported in frequency counts. I understand why this is the case, but it would be helpful if the authors presented a strategy. They are balancing a large sample for qualitative with representation.
Obviosly we didn’t manage to make our analysis method clear enough. Therefore we explained them and why we used this procedure in more detail.
The second point is that more details on who the participants were. People who volunteered to participate in this study, that is, with experience of after school activity might represent a specific sample. Children who avoid doing after school activities, for example, are not represented and yet, when start speaking of the health related benefits of doing after school activities, it is this group who would benefit the most. Rather than hide this issue in the article, my suggestion is that they express the sample as using people who volunteered and thus could have a positive bias from the outset.
We did give more information about the participants and how the sampling was made. Moreover, we mentioned the selection of the sample as a minor restriction in the limitations.
In summary, strengthen the methods and this should enhance the work.
Thank you for the productive feedback, we adapted the method section accordingly and are convinced that this strenghtend our paper.
Reviewer 2 Report
GENERAL COMMENTS
The aim of this paper was to examine students’ perspectives on extracurricular school sports trips on their involvement, active participation, and co-designing opportunities. Although this article addresses an interesting topic especially the post-covid19 pandemic, there are many major issues should be addressed before publication.
OVERALL COMMENTS
- References did not follow the journal’s format.
ABSTRACT
The objective of the manuscript is not clearly written. Suggest to rewrite it clearer.
What method used to analyse the qualitative text?
INTRODUCTION
The introduction needs major revision and clarification. First, the aim of the study is not clear. Qualitative research should answer a specific research question, which is lacking in this study. Also, the way the authors build up their introduction does not lead to the research question, significant of the study and contribution to knowledge. Although many of the necessary information regarding the background is already written down, the authors should re-structure their introduction, explaining why their research is important and what is really lacking. The authors described some studies on extracurricular school sports trips and health benefits separately, but the discussion of both together is very shallow. More importantly, this should lead to a clear research question.
METHODS
The methods section needs major revision. As it stands, it is not possible to replicate their study. The study need more justification. For example, why use small groups interview? How participants were selected? Will 12 years old’s motives to participate in extracurricular school sports trips will not be different from a 15 years old? Any struggle with language from the participants? Please attach the basic questions for the interview? Who did the analysis? Is the data achieve saturation? What method used to analyse the data? What is the procedure for data collection?
RESULTS
The results section needs major revision. The authors should add how they gathered all the information. Probably, including a research question would help the authors to structure their results. I think the authors put too many information in the tables, making the entire manuscript hard to follow. Table 1 should follow a review format – suggest deleting the type of reference and sources. Results illustration seemed not concise, keep on repeating similar information.
DISCUSSION
In the discussion section, the authors should further discuss their findings and the implication of these findings.
Conclusion is not concise and not concluding. Please revise.
Author Response
We want to thank the reviewer for the many helpful and thoughtful suggestions on how to improve the manuscript. Based on those, we formulated research questions and revised the other parts accordingly. Our adaptions (via track changes) will be explained according to the comments. We gladly addressed them as follows:
References did not follow the journal’s format.
Thanks for pointing this out, we now used the Zotero template of MDPI for the references.
ABSTRACT
The objective of the manuscript is not clearly written. Suggest to rewrite it clearer.
What method used to analyse the qualitative text?
We made a few adaptiations to the abstract to make the objective of the manuscript more clear. We already addressed the method of qualitative text analysis (Kuckartz, 2014) in the abstract.
INTRODUCTION
The introduction needs major revision and clarification. First, the aim of the study is not clear. Qualitative research should answer a specific research question, which is lacking in this study. Also, the way the authors build up their introduction does not lead to the research question, significant of the study and contribution to knowledge. Although many of the necessary information regarding the background is already written down, the authors should re-structure their introduction, explaining why their research is important and what is really lacking. The authors described some studies on extracurricular school sports trips and health benefits separately, but the discussion of both together is very shallow. More importantly, this should lead to a clear research question.
Now we adjusted the introduction based on the mentioned points, formulated research questions and brought up the issue of interaction effects in the limitations (outlook for future studies).
METHODS
The methods section needs major revision. As it stands, it is not possible to replicate their study. The study need more justification. For example, why use small groups interview? How participants were selected? Will 12 years old’s motives to participate in extracurricular school sports trips will not be different from a 15 years old? Any struggle with language from the participants? Please attach the basic questions for the interview? Who did the analysis? Is the data achieve saturation? What method used to analyse the data? What is the procedure for data collection?
Thank you for the critical look at our methods, which are now revised. We added the basic questions of each initial category to the manuscript and explained how we moved further. Furthermore we described the analysis in more details. Regarding the age differences there was only one student aged 12, two 16 and the rest 13 to 15 with a stable mean of 14 years. Our aim was to include students of two grades (leading to this age range), which are comperable in terms of their experience in participation in school sports trips.
RESULTS
The results section needs major revision. The authors should add how they gathered all the information. Probably, including a research question would help the authors to structure their results. I think the authors put too many information in the tables, making the entire manuscript hard to follow. Table 1 should follow a review format – suggest deleting the type of reference and sources. Results illustration seemed not concise, keep on repeating similar information.
We are thankful for the proactive idea with a concrete research question which we now mentioned in the introduction and used it throughout the manuscript. We agree that table 1 is quite large, but we think it is important in terms of comprehensibility which the reviewer rightly remarked in the upper comment. Moreover, we went through the results and regrouped some categories, which should now be more concise.
DISCUSSION
In the discussion section, the authors should further discuss their findings and the implication of these findings.
We rearranged parts of the discussion based on the research questions and did some minor adjustments.
Conclusion is not concise and not concluding. Please revise."
We agree that some points of the conclusion are more related to general recommendations instead of concluding arguments. Therefore we changed this accordingly.
Round 2
Reviewer 2 Report
Well done, the revised version showed a lot clearer and thus I recommend to accept the manuscript.